# Time of Day Analysis over a Field Grown Developmental Time Course in Rice

**DOI:** 10.3390/plants12010166

**Published:** 2022-12-30

**Authors:** Todd P. Michael

**Affiliations:** The Plant Molecular and Cellular Biology Laboratory, The Salk Institute for Biological Studies, La Jolla, CA 92037, USA; tmichael@salk.edu

**Keywords:** diurnal, circadian, development, *Oryza sativa*, field conditions

## Abstract

Plants integrate time of day (TOD) information over an entire season to ensure optimal growth, flowering time, and grain fill. However, most TOD expression studies have focused on a limited number of combinations of daylength and temperature under laboratory conditions. Here, an *Oryza sativa* (rice) expression study that followed TOD expression in the field over an entire growing season was re-analyzed. Similar to *Arabidopsis thaliana*, almost all rice genes have a TOD-specific expression over the developmental time course. As has been suggested in other grasses, thermocycles were a stronger cue for TOD expression than the photocycles over the growing season. All the core circadian clock genes display consistent TOD expression over the season with the interesting exception that the two grass paralogs of *EARLY FLOWERING 3* (*ELF3*) display a distinct phasing based on the interaction between thermo- and photo-cycles. The dataset also revealed how specific pathways are modulated to distinct TOD over the season consistent with the changing biology. The data presented here provide a resource for researchers to study how TOD expression changes under natural conditions over a developmental time course, which will guide approaches to engineer more resilient and prolific crops.

## 1. Introduction

Plants have evolved an internal timing mechanism called the circadian (*circa diem*) clock that maintains an about 24 h period but can vary by several hours to ensure proper synchronization with external seasonal and latitude associated changes in light and temperature [1]. The circadian clock is conserved at the level of genes and molecular architecture across the green lineage from single cell alga to higher plants [2]. The circadian clock interacts with the predominant external timing cues of light and temperature to partition or phase the expression of almost all genes to specific time of day (TOD) to ensure proper growth and development [3,4]. However, most studies only look at a limited number of lab-based conditions to study the TOD regulation of gene expression, while there is a need to better understand TOD regulation under natural and field conditions over development [5,6,7].

There are a growing number of studies cataloging the TOD regulated genes across many species under standard lab conditions at one or a limited number of developmental time points [3,8,9,10,11,12,13,14,15,16,17,18,19]. In general, greater than 30% of genes cycle under driven conditions of light and/or temperature cycles, while between 10–20% genes cycle under continuous conditions that enable the detection of genes specifically controlled by the circadian clock. There are exceptions like the non-grass monocot *Wolffia australiana*, that has a minimal body plan, fast life cycle and only 13% of its genes cycle in a TOD fashion under light cycles [15]. In *Arabidopsis thaliana*, when eleven different conditions of light, temperature and daylength were tested, greater than 90% of genes displayed TOD expression under at least one condition, consistent with different environmental conditions playing specific roles in phasing biological processes [3].

Plants under natural conditions experience changing day length and thermocycles that they must integrate to synchronize with the environment over time [3,7]. Over the year, the day length and solar radiation vary by latitude with the longest and shortest days (solstice) occurring in June and December, respectively, while temperature also varies by magnitude between night and day and overall average highs and lows, yet with a lag in reference to the solstices. Over the day, light and temperature also vary with temperature lagging the increasing solar radiation, and this relationship changes by time of year as well as latitude (Appendix A). In fact, both natural and selected plants have circadian period lengths that are correlated with their environment [1,20,21]. The ability to accurately synchronize with the external environment confers a fitness advantage by enabling the anticipation of dawn and dusk under the changing daily conditions [1,22,23].

The basic architecture of the core circadian clock is conserved across the green lineage, although in higher plants there is considerable overlap with light signaling and flowering time genes [2]. At the core, the circadian clock is a complex network of negative and positive feedback loops based on myeloblastosis (MYB) transcription factors (TFs) *LATE ELONGATED HYPOCOTYL* (*LHY*) and *CIRCADIAN CLOCK ASSOCIATED 1* (*CCA1*) and *PSEUDO RESPONSE REGULATOR* (*PRR*) gene family [24]. *LHY* and *CCA1*, and the related *REVEILLE* (*RVE*) family, are exclusively expressed around dawn, while the *PRR* family display “circadian waves of expression” spanning dawn (*PRR9*), midday (*PRR7*), evening (*PRR3* and *PRR5*) and early night (*PRR1*) [25]. The *MYBs* and *PRRs* are genetically linked in the genome suggesting there is evolutionary pressure to ensure the feedback back loop is intact to retain strain-specific cycling; *PHYTOCHROME INTERACTING FACTOR 3* (*PIF3*) and *PHYTOCHROME A* (*PHYA*) are also genetically linked over evolutionary time suggesting that there is pressure to maintain the light regulated growth pathway [2]. There is a tight association between light signaling, temperature sensing and flowering time since genes like *EARLY FLOWERING 3* (*ELF3*) and *PHYB* have been shown to play roles in each process and disrupt the circadian clock [26,27]. However, in *Arabidopsis* there are at least two independent circadian mechanisms where one is light dependent and the other is temperature dependent, suggesting that while these signals are integrated, they also can act independently [28].

As an example of how the core circadian clock genes are critical for the integration and optimization of daily and seasonal integration of the environment, there is a growing literature that clock, light and flowering time genes are some of the major targets of modern breeding programs [29,30,31]. In *Glycine max* (soybean) it has been shown through quantitative trait loci (QTL) and genome wide association studies (GWAS) that *PRR3* impacts flowering time and is the target of domestication [32,33]. *PHYA* has been identified as the maturity group (MG) loci E3 and E4 in soybean, and it is formally possible that the *PHYA*-*PIF3* linkage is responsible for the environment-specific nature of the E3 and E4 loci [2,34]. In the grass monocots, *PRR3* and *PRR7* are referred to as *PRR37* and *PRR73*; however, with the increased number of high-quality monocots and sister lineage genomes, it is possible to assign the define these *PRRs* based on synteny [2]. This distinction is important due to the fact that in *Sorghum bicolor*, *SbPRR37*, which is more closely related to *PRR3*, was identified as a maturity associated (ma) locus involved in flowering time [35]. while in contrast, the rice *OsPRR37*, also known as *heading date 2* (*hd2*), is more closely related to *PRR7* and provides photoperiod and temperature sensitivity [36]. A clear understanding of how these genes impact and improve agronomic characteristics over a growing season is still needed.

Very few studies have looked at TOD expression over a complete developmental time course in a crop [5]. One study looked at TOD expression from seedling to harvest in rice (*Oryza sativa japonica* nipponbare) using microarrays over nine developmental time points spanning the growing season in Tsukuba, Japan (Appendix A) and sampling every two hours for 48 h [37,38,39,40]. Initial analysis of this dataset revealed gross developmental differences in the vegetative, reproductive, and ripening phases [40]. Leveraging modeling approaches, two studies of this dataset showed that temperature, plant age and the diurnal rhythms play a dominant role in establishing expression patterns in the field and that the internal timing of the plants is relatively constant [37,39]. However, these studies have not published a full TOD analysis of the cycling patterns across development to identify changes in biological phasing during a field season. Therefore, the time course was re-analyzed through the DIURNAL and ELEMENT [3,41] TOD analysis pipeline to identify specific cycling parameters, significant cis-elements and biological processes as defined by gene ontology (GO) terms.

## 2. Results

### 2.1. Identification of Time of Day (TOD) Expression in the Rice Developmental Time Course

The RiceXPro field grown time course is a rich dataset to identify changing TOD expression patterns at different developmental stages. The eleven time courses, nine from leaf tissue and two from root tissue, spanning the entire growth season in Tsukuba, Japan (Appendix A), were processed through the DIURNAL pipeline to identify the changes in TOD expression over the growing season [3]. Models were developed for HAYSTACK to identify cycling genes from the time courses that were collected every two hours over 48 h. There was a total of 27,648 genes on the arrays and all genes were expressed under at least one of the eleven conditions; there was a range of between 6% (root 15 d) and 21% (leaf 99 d) of genes not expressed (Table 1; Appendix A). Significant cycling genes were identified from expressed genes resulting with leaf genes ranging from 57% (leaf 29 d) to 88% (leaf 88 d), and root genes were much lower at 9% (root 15 d) and 5% (root 43 d) (Table 1; Appendix A). There were 3689 (13%) genes that were neither expressed nor rhythmic across all time courses (Appendix A), while there were only 163 genes that were expressed yet not called significantly cycling (Appendix A). There were 378 genes that cycled under all eleven conditions (Appendix A), 5929 cycled in all leaf but not in root (Appendix A), and all genes that cycled under at least one leaf condition cycled in roots.

Consistent with the lower number of genes cycling in the roots, the leaf and root expression was poorly correlated in contrast to most leaf samples (Figure 1A and Appendix A). Looking at the samples in a multi-dimensional scaling (MDS) plot, revealed that the leaf samples followed a circular progression suggesting early vegetative phases (15 and 29 d) were most similar to the ripening phase (113 and 92 d) (Figure 1B). Both day length and temperature vary over the developmental time course, and temperature is relatively similar between 15 d and 113 d, which could be influencing the circular MDS plot (Appendix A). Root samples were set apart consistent with the correlation plot yet proximal to the same developmental time as the leaf samples. In general, there was a high overlap between the genes that cycle per condition as evidenced for the first five developmental times, yet there were many genes that are specific as well (Figure 1C and Appendix A). Since 15 and 78 d were the furthest apart of the MDS plot, compared to 113 d, it seemed there should be a large difference in the genes that cycled between the three conditions, yet there was a similar overlap in cycling genes (Figure 1D,E). Therefore, since a high overlap between conditions was observed, it could be that phase, or expression TOD, was driving the relationships between time courses.

### 2.2. The TOD of Expression Changes over Developmental Time

All the developmental time courses showed the typical increase in the number of cycling genes at dawn and dusk (Figure 2A); however, several were less pronounced than others (Appendix A). Since the time points were set to the same time of day but the timing of sunrise and sunset changed over the time course (Appendix A), it was expected that there would be a phase shift of some genes in relation to the collected time points (Appendix A). Leveraging 15 d as an anchor point (baseline phase analysis), it was observed that a set of genes maintained the same phase across the entire time course (~17 to 27%), while most genes shifted progressively later, consistent with the later timing of temperature (Figure 2B and Appendix A). Time point 43 d on 19 June was the longest day of the time course, with dawn getting progressively later, and dusk progressively earlier after 3 July. In contrast, the temperature peaked in late august, with the warmest day occurring between 14 August and 21 August (Appendix A). If the photocycles were primarily driving the phase of expression, then the maximum difference in phase would occur in June. The fact that the phase shifts were more pronounced in 85 and 92 d suggested that thermocycles or absolute temperature played a more prominent role in setting the phase of expression; a previous modeling analysis found similar dependence on temperature [37,39]. In addition, it has been shown that thermocycles play a prominent role in setting TOD expression in *Brachypodium distachyon* [10].

The baseline phase analysis suggested that thermocycles could be a stronger cue for TOD expression. However, it could be that there was an artificial phase difference by only using 15 d as an anchor point. Therefore, an incremental phase shift analysis was carried out that looked at pairwise phase changes between proximal time points. If there were only an incremental change due to continuous changes in light or temperature the phase differences would be constant across developmental phases in the incremental phase analysis. However, an increase was observed in phase differences between 78 and 85 d as well as a more pronounced difference between 85 and 92 d (Figure 2C). In fact, the genes that displayed a phase difference between 6 and 19 h showed a predominant shift from afternoon to mid-evening expression with gene ontology (GO) terms all in the cellular component category associated with protein, endomembrane, golgi and respiration (Figure 2D,E). The pronounced phase change was not seen in the next comparisons (92 vs. 99 d and 99 vs. 113 d), consistent with the phase shift stabilizing in later developmental stages, which possibly was associated with the shift from reproductive to ripening phase.

### 2.3. The Core Circadian Clock and Flowering Time Genes over Developmental Time

The molecular mechanisms of the plant circadian clock were worked in the model plant *Arabidopsis thaliana*, and now there is a growing body of research demonstrating that not only is the circadian clock conserved at the level of genes and TOD expression, but also in overall organization [2]. The first plant clock gene identified was *timing of cab expression 1* (*toc1*), which was later cloned and shown to encode *PSEUDO-RESPONSE REGULATOR 1* (*PRR1*) with conserved CCT (CONSTANS, CO-like, and TOC1) and REC (receiver) domains [42]. *TOC1/PRR1* is part of a five gene family including *PRR3*, *PRR5*, *PRR7*, and *PRR9* that is regulated in “circadian waves of expression” with peak gene expression at 13, 11, 8, 7 and 4 h (hrs) after lights on (dawn), respectively [3,25]. In contrast to *Arabidopsis*, the entire *PRR* family in rice was expressed around the same TOD in the evening across the entire developmental time course (Figure 3; Appendix A), which was consistent with previous studies across an array of environmental conditions in rice [8] and other grasses [18].

The second family of core circadian clock genes identified were *LATE ELONGATED HYPOCOTYL* (*LHY*) and *CIRCADIAN CLOCK ASSOCIATED 1* (*CCA1*), which are part of a sub-family of the large myeloblastosis (MYB) transcription factors (TF) family defined by the “SHAQKYF” protein motif (*sMYB*) [43,44]. Both *LHY* and *CCA1* have peak expression at dawn in *Arabidopsis* regardless of entraining conditions (different photocycles and thermocycles) or under diurnal or circadian conditions [3]. Rice only has *LHY*, since *CCA1* is specific to the eudicot lineage [2], and its expression also tracks dawn regardless of the change in development or external conditions (Figure 3B; Appendix A). In addition to *LHY*, the sMYB sub-family consists of eight other genes that share both the “SHAQKYF” motif as well as dawn-specific expression and thus were named *REVEILLE* (*RVE*) [45], which also were expressed at dawn without much variation across the time course (Appendix A). In general, the core circadian clock genes did not alter phase appreciably over the developmental time course suggesting there may be downstream components of the light signaling or flowering pathway such as *GIGANTEA* (*GI*) (Figure 3E) that plays a role in diurnal or circadian expression that is sensitive to the changing daylength and temperature cycles.

### 2.4. ELF3 Is a Photo and Thermocycle Integrator over Developmental Time

In a previous study, the interaction between light and temperature was explored in the TOD expression in rice [8]. It was found that 54% (2415) of genes cycling under both light entrainment only (light/dark/hot/hot; LDHH) and temperature entrainment only (light/light/hot/cold; LLHC) displayed shifted or anti-phasic (6–12 h) expression, in contrast to 14% having shifted expression between LDHH and light and temperature entertainment (light/dark/hot/cold; LDHC) or LLHC and LDHC. These results suggest that light and temperature have distinct impacts on the TOD expression of a sub-class of genes (Appendix A). Of these genes that display anti-phase expression between light and temperature were multiple important clock and light associated genes such as *DE-ETIOLATED 1* (*DET1*), *PHYTOCHROME A* (*PHYA*), *PHYC*, *CRYPTOCHROME 3* (*CRY3*), *EARLY FLOWERING 3* (*ELF3a*), and *ELF3b*. *PHYA* had very focused TOD expression after dusk at the beginning of the developmental time course, and then its peak expression expanded and was higher amplitude with an earlier phase (Figure 3F). One interpretation of this pattern of expression was that light and temperature played differing roles through development to ensure correct TOD phase of *PHYA*.

There are two paralogs of *ELF3* that both have anti–phasic expression between LDHH and LLHC; in each condition, the two paralogs have the same TOD expression (Appendix A). However, in the developmental time course they had anti-phasic expression with each other (Figure 3G,H), which was similar to the phasing of these two genes under light and temperature entrainment (LDHC) and in other grasses (Appendix A) [8,18]. These results suggested that the two paralogs were sensitive to both light and temperature, and maybe their timing relative to one another provided seasonal acuity. This was consistent with *ELF3* playing a role in temperature sensing and thermoregulated growth [26,46,47,48,49]. Moreover, *ELF3a*, which was initially identified as *early flower 7* (*ef7*) and *heading date 17* in rice [50,51], also had higher expression during the earlier part of the developmental time course (vegetative and reproductive), while *ELF3b/ef3* had higher amplitude expression during the reproductive phase (Figure 3G,H). *ELF3* is part of the evening complex (EC) along with *LUX ARRHYTHMO* (*LUX*) and *EARLY FLOWER 4* (*ELF4*), and it was recently shown that the two *ELF3* paralogs may represent two separate ECs and that the EC is required for photoperiodic flowering [52,53]. Therefore, it is possible that the two *ELF3* paralogs represent EC that cover different developmental phases; consistent with this *ELF4b* and *ELF4c* were only rhythmic during the late reproductive phase and the ripening phase (Appendix A).

### 2.5. Element Analysis over the Developmental Time Course

The high level of coordinated TOD expression enables the identification putative cis-acting elements that provides further clues as to the gene networks. Diurnal and circadian elements are summarized into the morning (Morning Element, ME; GBOX), evening (Evening Element, EE; CCA1 binding site, CBS; GATA), and midnight (Protein Box, PBX; Telo Box, TBX; Starch Box, SBX) modules as central to TOD regulation, which are conserved across the green lineage [3,8,15,17,54]. The ELEMENT pipeline takes a list of coordinated genes and searches for 3–8 bp kmers in promoter regions (500 bp) that are overrepresented against a dictionary of all possible 3–8 bp kmers across the dataset [3]. There were 253 lists (11 time courses and 23 time points) of genes with a specific TOD expression, and in total ELEMENT found 1749 3–8 kmers that were significant (*p* > 0.05), which was summarized based on overlapping sequence into 157 motifs. On a per time course basis, the root time courses had very few elements with only 9 and 12 significant 3–8 bp kmers for 15 d and 43 d, respectively, which summarized into the EE (AATATCT). In contrast, for leaf the number of significant 3–8 bp kmers varied between 134 and 907, while the clusters varied between 28 and 88 (Appendix A). The number of significant 3–8 bp kmers increased from vegetative to the reproductive phase peaking at 78 and 85 d, which based on the methodology of how the elements were identified suggested that the networks were more coordinated during the reproductive and ripening phases.

All the elements that have been identified before in rice and other species were identified across the developmental time course in addition to novel elements that were specific to TOD and develop (Appendix A). Since all 3–8 bp kmers were tested for significance across each gene list, it was possible to plot the TOD that any given element was significant, such as the core elements ME, EE, CBS and the TBX that were significant at different TOD (Figure 4). These core elements maintained a similar phase of overrepresentation across the entire time course consistent with circadian clock maintaining a similar relative phase despite the changing light and temperature changes. Using modeling techniques on this dataset it was found that the overall internal timing only shifts by 22 min, which is less than the resolution of this analysis [39]. There were many developmental and TOD-specific elements that may explain the shifts in phase and expression of downstream networks, which could be useful as targets to modify specific pathways. In both rice and *Arabidopsis*, the TBX shifts TOD overrepresentation to evening-specific under light cycles alone (LDHH), while it is midnight-specific under temperature cycles alone (LLHC); in this dataset the TBX is overrepresented around midnight suggesting that the temperature cycles have a stronger influence on the pathways involving the TBX.

### 2.6. Gene Ontology (GO) Analysis Reveals Rewiring of the Plant over Development

While it is assumed that TOD regulation changes over development, it is not known how the timing of specific pathways changes over the entire growing season. Gene ontology (GO) overrepresentation was used to estimate changes in the expression pathways over the developmental time course. The number of significant (FDR < 0.05) GO terms over the entire developmental time course was 870 (out of 11,290), with each time course having between 146 (29 d) and 435 (85 d) significant GO terms at one TOD (Figure 5A, Appendix A). There were 46 significant GO terms that were shared across all developmental time courses, and an incremental analysis looking at the developmental time courses in a sliding window revealed an increasing overlap in significant GO terms (Figure 5B, Appendix A). Based on developmental phases, vegetative was distinct from both reproductive and ripening, which were more alike (Appendix A).

The timing over the day of the significant GO terms provides information as to the phase of the biological processes. Most (55–71%) GO terms were only significant at one hour over the day, while some time courses (43 d, 78 d, 85 d and 99 d) showed progressively more GO terms significant at several times over the day consistent with broader regulation of that pathway or process (Figure 5A). The higher number of GO terms with broader expression supported the increased number of elements identified at these developmental time points and suggested the samples taken at these time points were more coordinated. If only the GO terms with more than one occurrence over the day were ranked, the vegetative phase was dominated by GO terms like chloroplast, plastid and processes associated with photosynthesis, while RNA metabolic process and nitrogen compound metabolic process were several that had higher ranking (more occurrences per day) during the reproductive and ripening phases (Appendix A).

Looking at the TOD significance provides clues as to the timing as well as shifts of the biology over the developmental time course. Several representative GO terms that were significant at specific times over the developmental time course were plotted and the full dataset is available (Figure 5; Appendix A). The GO terms associated “plastid” were significant at dawn into the early day and display a unique pattern with the overrepresentation starting earlier by 78 d (Figure 5C); this pattern suggested that developmental age played a role in the timing of the plastid function since it was opposite to the timing of external daylength and temperature. In contrast, the GO term “nitrogen metabolic processes” was significant at dawn and stable across the developmental time course (Figure 5D). The GO term “response to heat” was an example that was mostly specific to the reproductive phase while “magnesium chelatase complex” was an example of a term that was only significant one time of day across all developmental time points (Figure 5E,F). There were also terms like “protein binding” that were mostly specific to a TOD, in this case just before dusk, but were also found at other TOD (Figure 5H). Finally, there were examples like “exocyst” which were specific to a developmental phase, in this case vegetative, and a TOD such as midday (Figure 5I). Overall, these results suggested that rice focused more energy on making proteins as the plant aged, but this could potentially have reflected a limitation of the study that it only looked at leaf material.

## 3. Discussion

Few studies have evaluated the changes in time of day (TOD) expression over a complete growing season and development cycle to identify the interactions between plant age, seasonal changes, and stochastic changes in the field. Here, a previously published rice dataset that measured expression levels every two hours over 48 h at nine developmental time points spanning the four-month growing season and two tissue types was extended with a full TOD analysis [38]. The nine leaf time courses are equally distributed over three developmental phases (vegetative, reproductive, and ripening), which are more similar to one another than across developmental phases, although the late ripening time point (113 d) had similarity to the first vegetative time point (15 d) (Figure 1). Very few genes are predicted to cycle in the roots, yet the genes that were called cycling, also were predicted to cycle in all the leaf time courses. Both a baseline and incremental phase shift analysis showed that temperature plays a dominant role in setting TOD expression for a subset of genes associated with protein, endomembrane, golgi and respiration (Figure 2). In contrast, the core clock component family of *PSEUDEO-RESPONSE REGULATORS* (*PRRs*) maintains a stable TOD expression over the entire developmental time course like the overrepresentation of the highly conserved Evening Element (EE), CCA1 binding site (CBS), Morning Element (ME) and Telobox (TBX) cis-elements (Figure 3 and Figure 4). *EARLY FLOWERING 3* (*ELF3*), which is thought to play a role in temperature sensing and has been implicated in photoperiodic flowering, has two paralogs that display anti-phase expression under natural conditions of both light and temperature, and are expressed differentially over development (Figure 3). In addition to the shift in TOD expression, there is an increase in both significant cis-elements and gene ontology (GO) terms around the warmest days in August and transition between reproductive to ripening (85/92 d), suggesting a more focused TOD regulation over development (Figure 4 and Figure 5). The genes, cis-elements, and GO terms identified per developmental time/plant age and seasonal environment are complementary to the previous modeling efforts on this dataset [37,39], and provide a resource for future breeding and engineering efforts in rice.

In the initial study of this development dataset, the authors only report that 7% of the samples showed diurnal patterns of cycling [40]. Based on the description in the text, the authors may have only compared a daytime (12 p.m.) versus midnight (12 a.m.) for samples grouped by developmental phase. In contrast, with a full DIURNAL analysis for each of the separate time courses here, almost all genes classified as expressed cycle under at least one condition (Table 1). In rice, there have been many global time courses under lab or few environmental conditions that have reported between 10% to 60% cycling under diurnal conditions [55,56,57,58,59]. Each of these studies uses a slightly different sampling approach, tissue type and software to identify cycling genes, which could explain the variation in the overall number of cycling transcripts. In a study that looked at both diurnal and circadian conditions in rice and compared entrainment with light (light/dark/hot/hot; LDHH), temperature (light/light/hot/cold; LLHC) or both (light/dark/hot/cold; LDHC), found that greater than 60% of genes cycled under at least one condition, consistent with the fact that some genes cycle under specific environmental conditions [8]. A similar study design in *Arabidopsis* that tested eleven diurnal and circadian conditions, which also included continuous dark and day length changes, found greater than 91% of expressed genes cycled [3]. Both studies were in lab conditions with the temperature in phase with the abrupt light/dark transitions. Therefore, the high number of cycling genes in rice across development may reflect the role of light, temperature, and other untested entraining signals (Appendix A) that may play a role in driving TOD expression.

Multiple studies leveraged this developmental time course to model the effects of the field grown environment and have found that temperature had a strong impact on gene expression while the internal phase of the plant stays constant (within 22 min) despite changing photoperiod, temperature, and developmental age [37,39]. Consistent with these findings, the shift in TOD expression tracks the increasing temperature over the season, and less so the change in photoperiod (Figure 2). Moreover, genes that showed the largest TOD shift during the critical temperature period had similar GO terms as those preferentially phased by temperature cycles alone (LLHC) [3,8]. It has been shown in *Brachypodium distachyon* that thermocycles entrain leaf movement over other stimuli [60]. In fact, time courses across light and temperature entrainment regimes demonstrated that thermocycles play a dominant role in setting the phase in Brachypodium [10]. However, similar to the modeling results [39], some of the core clock genes and their cognate cis-elements do not shift TOD over development, confirming that the internal timing is relatively constant while external cues like temperature may play a role in shaping specific aspects of biology (Figure 3 and Figure 4). It is possible that temperature playing a predominant role in TOD regulation is a common feature of grass monocots since they emerged in the shady understory where thermocycles may have played a more predominant role than light [2].

Recently it has been shown that *ELF3* plays a role in temperature sensing [26,46,47,48,49]. Rice *ELF3a* and *ELFb* have the same phase of expression as each other under light (LDHH) and temperature (LLHC) entrainment, yet they are anti-phasic with dawn and dusk expression, respectively (Appendix A) [8,41]. However, when entrained under light and temperature (LDHH), the phase of expression of these two genes moves from in-phase to anti-phase [8,41], like the profile seen under natural conditions in this dataset (Figure 3) and across other grasses (Appendix A). There may be genetic component to this expression pattern since the restorer line Minghui86 that is used to make the super hybrid Liangyou2186, has both paralogs in phase with one another while they are anti-phase in the male-sterile line SE21s and the hybrid Liangyou2186 (LY2186) and its parental lines including the male-sterile line SE21s and the restorer line Minghui86 (MH86) (Appendix A) [57]. These results show that the *ELF3* paralogs are differentially sensitive to light and temperature input, and that under natural conditions, their expression is maintained in anti-phase, which could provide information to the plant as to the interplay of light and temperature over the season. Recently it was shown through knockout and interaction studies that the evening complex (EC) components *ELF3*, *LUX ARRHYTHMO* (*LUX*) and *EARLY FLOWER 4* (*ELF4*) are required for photoperiodic flowering and that they form two ECs with the more dominant one having *ELF3a/ELF3-1/HD17* [52,53]. Based on the developmental time course, the amplitude of *ELF3a* is stronger in the vegetative and reproductive, while the expression decreases or becomes arrhythmic in ripening; in contrast, *ELF3b* increases amplitude during the ripening phase (Figure 3). These results suggest that *ELF3* integrates light and temperature not only over the day but also developmentally and the two ECs may play distinct roles in controlling photoperiodic flowering.

One of the driving forces to re-analyze these data was to identify the biological networks that change by season. Since TOD expression analysis captures the timing of many biological processes over the day, it is possible to see how the processes change across the plant by season. Previously it was found that the ripening phase represents a shift in the transcriptome, but also it was shown that the ripening phase was also similar to the vegetative phase [40]. Similarly, the TOD level analysis revealed that the vegetative time points were also like the ripening time points while overall expression clearly separated the time courses into the three phases (Figure 1). However, the largest phase shift occurred at the reproductive to ripening transition (85 d to 92 d) where a subset of genes shifted phase greater than 6 h and GO term overrepresentation analysis revealed protein, endomembrane, golgi and respiration. In addition, the number of significant GO terms and the occurrences of those GO terms increased over development with the most at the same time as the maximal phase shift and also the highest number of significant cis-elements (Figure 2, Figure 4 and Figure 5). One interpretation of these results is that TOD regulation is more coordinated during this developmental phase (reproductive to ripening phase), possibly an important timing for grain yield. However, an alternative explanation is that the tissue sampled at the later time points when the plants are larger represents a smaller, less diverse sample than that taken when the plant is younger, and the sample represents a higher percentage of the plant. Tissue, cell, and TOD-specific experiments will be required to differentiate these possibilities and to gain a full understanding of changing biological processes over the season [60].

In the day and age of advanced breeding using editing strategies such as CRISPR-CAS9 there is a growing need to fully understand the gene networks over developmental and TOD, especially as it is becoming clear that circadian, light, and flowering genes are a major target of breeding programs [31]. A very nice study compared the use of an advanced chamber to simulate field conditions and found that it was possible to recapitulate TOD cycling at the one plant age tested, yet there were limitations in light quality (red/far-red, UV) and stochastic stressors that can be missed in these systems [61]. However, what was missing in this analysis was the concept of plant age and tissue, which both require a deep understanding to edit plants of the future. For instance, it was found in sugarcane that looking at two developmental time courses at four and nine months revealed changes in TOD expression like what is observed in this study for some classes of genes; in addition, they show that microclimates, and by extension, the tissue sampled impacts the TOD expression [62]. Moreover, several recent studies have shown that the TOD expression can differ by cell type, which suggests that some features of biology could be manipulated in a cell-specific manner [31]. Future studies will need to include plant age, field conditions and TOD single cell to fully explore the complexities of the plant response in the field.

## 4. Materials and Methods

### 4.1. Developmental Rice Time Course

The rice (*Oryza sativa* var nipponbare) developmental time course was accessed on the RiceXPro web site (https://ricexpro.dna.affrc.go.jp/; accessed on 9 September 2020); it is also available from the Gene Expression Ominbus (GEO): GSE36040 (leaf) and GSE39423 (root). The data have been described in several publications [13,37,38,39] and the leaf and root diurnal time courses are briefly described here. Leaf (vascular leaf; flag leaves) and root tissue were collected over two days at eleven different developmental stages from field grown plants in Tsukuba, Ibaraki, Japan (36.0835° N, 140.0764° E) between 3 June 2008 and 11 September 2008 (Appendix A). The eleven individual time courses were collected every two hours over three days starting at 10:00 am local time spanning Zeitgeber Time 6 (ZT06) hrs to ZT54 hrs. Sunrise (dawn) and sunset (dusk) fluctuate over the time course so 10:00 am was used as a set point for analysis. Daylength changed from 14:30 h of light in June with temperature fluctuation of 16–26 °C to 12:30 h of light in September with temperature fluctuation of 18–28 °C; peak day length was at the end of June, while peak temperature in August at 32 °C (Appendix A).

### 4.2. Time Course Analysis

Data were downloaded from GEO GSE36040 (leaf) and GSE39423 (root) and processed using the DIURNAL pipeline described in other studies [3,8,10,15]. Briefly, several different types of normalization were tested on the time courses to achieve the best results; ultimately quantile normalization was leveraged. Normalized data were then used to identify cycling genes using HAYSTACK, which fits predefined models to the data to identify genes with cycling behavior. Several tools were also tested to compare results (metacycle), and HAYSTACK results using a previously empirically defined R > 0.8 was used to identify significantly cycling genes; time of day (TOD) expression is assigned as the time of peak expression in hrs relative to the sample collection (10:00 am). Since all the time courses were started at 10:00 am, and dawn shifted by up to an hour over the period of the developmental collection, 10:00 am was defined as ZT0. The complete time course analysis (Appendix A), TOD phase summary (Appendix A), and Summary Table (Table 1) are included.

### 4.3. Circadian Clock Ortholog Analysis

Rice core circadian clock orthologs genes were identified using known *Arabidopsis thaliana* genes [2], and publications on the rice circadian clock [8,63]. Orthologous genes were identified across other model and crop species with time courses using orthofinder [64]. Protein models were downloaded for *Zea mays*, *Sorghum bicolor*, *Brachypodium distachyon* and *Setaria italica* from Phytozome (https://phytozome-next.jgi.doe.gov/; accessed on 5 May 2020) [65].

### 4.4. Analysis of Other Time Courses

The rice developmental time courses were compared to previous time courses of rice under different light and temperature conditions [8], rice hybrids [57] and maize, *Sorghum* and *Setaria* [18]. The expression data were downloaded from DIURNAL [41] for the rice under light and temperature conditions; the rice hybrid dataset (GSE138193) was downloaded from NCBI GEO; and the grass dataset was downloaded from the journal web site [18]. The analysis and normalization for each was used for the comparisons.

### 4.5. ELEMENT Time of Day (TOD) Cis-Element Identification

Once cycling genes were identified, putative cis-acting elements associated with TOD expression were identified using the ELEMENT pipeline [3,8,15,17]. Briefly, promoters were defined as 500 bp upstream of genes, and were extracted to compute background statistics for all 3–8 bp kmers. Promoters for cycling genes were split according to their associated phase and 3–8 bp kmers that were overrepresented in any of these 23 promoter sets were identified by ELEMENT. By splitting up cycling genes according to their associated phase (253 lists), power to identify 3–8 bp kmers associated with TOD-specific cycling behavior at every hour over the day was achieved. The threshold for identifying a 3–8 bp kmers as being associated with cycling was an FDR less than 0.05 in at least one of the comparisons. The significant 3–8 bp kmers were automatically clustered according to sequence similarity into motifs. Sequence logos were generated from the 3–8 bp kmers matrix associated with each summarized motif.

### 4.6. Gene Ontology (GO) Analysis

The array data were based on The International Rice Genome Sequencing Project (IRGSP) Releases Build 4.0 of the rice (*Oryza sativa* ssp. japonica cultivar Nipponbare) genome. Therefore, the annotation was updated for the Build 4.0 rice gene predictions using EggNOG-mapper [66]. GO terms were extracted from the new annotation for GO term analysis using GOATOOLS [67]. Significant (*p* > 0.05) GO terms were identified by looking at all 253 phase-specific gene lists with the flags (—pval = 0.05 —method = fdr_bh —pval_field = fdr_bh). All GO term associations were identified by setting the pval flag to 1.

## Figures and Tables

**Figure 1 plants-12-00166-f001:**
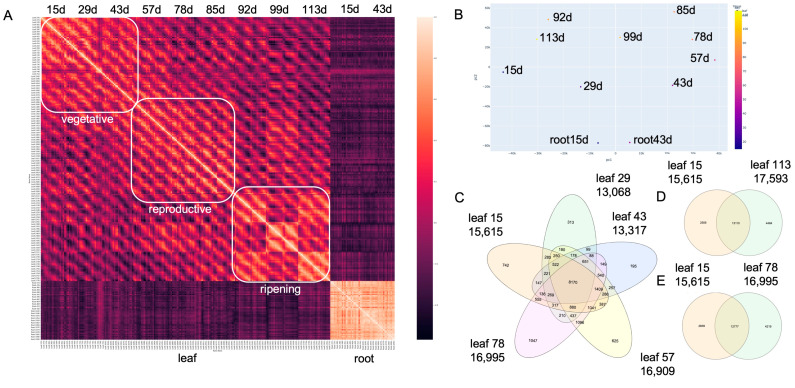
DIURNAL analysis of Developmental time course in rice. (**A**) Correlation matrix of the full rice developmental time course in order from 15 to 113 day (d) and from Zeitgeber Time 6 (ZT6) to ZT54 from leaf, and then 15 and 43 d from root; (**B**) Multidimensional scaling plot of the time courses; (**C**) Overlap of cycling genes from 15 to 78 d spanning vegetative and reproductive phases; (**D**) Venn diagram of leaf 15 d versus leaf 29 d; (**E**) Venn diagram of leaf 15 d versus leaf 113 d.

**Figure 2 plants-12-00166-f002:**
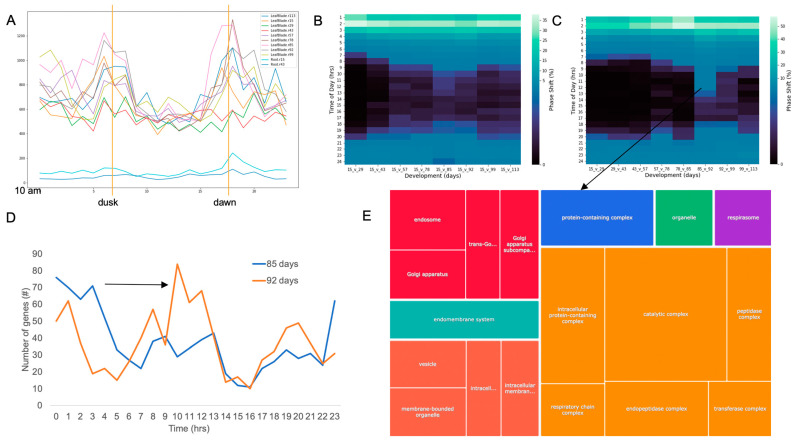
Global phase shifts associated with developmental timing. (**A**) Number of genes with specific time of day (TOD) expression across the eleven time courses. The time courses all started at 10 am, so relative dusk and dawn were marked; (**B**) relative phase shift compared to the first time course at 15 d with hrs shifted on the *y*-axis and the comparison time courses on the *x*-axis; (**C**) incremental phase shift comparing each proximal time courses with hrs shifted on the *y*-axis and the comparison time courses on the *x*-axis; (**D**) The phase shift of the genes from the incremental phase analysis between 85 and 92 d that showed a predominant shift from afternoon to mid-evening expression; (**E**) The significant gene ontology (GO) terms for the genes displaying the phase shift from the incremental phase analysis between 85 and 92 d; colors are arbitrarily assigned to GO terms with similar classifications.

**Figure 3 plants-12-00166-f003:**
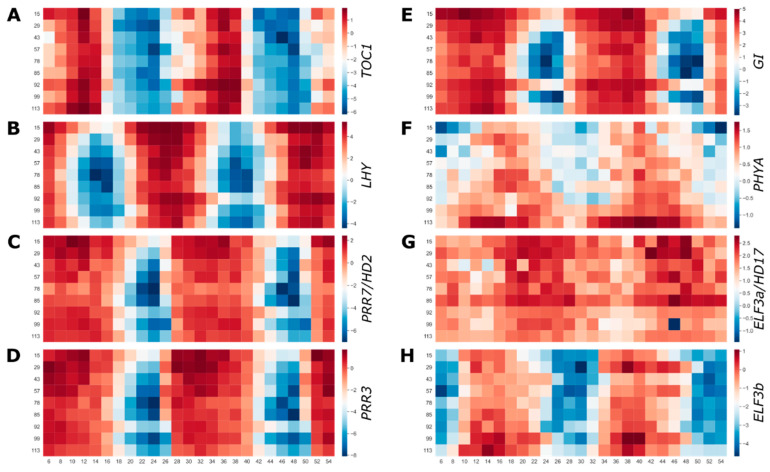
Core circadian clock, light and flowering genes across the developmental time course. Gene expression across the developmental time course in leaves for (**A**) *TIMING of CAB EXPRESION 1* (*TOC1*)/*Pseudo-response regulator 1* (*PRR1*); (**B**) *LATE ELONGATED HYPOCOTYL* (*LHY*); (**C**) *PRR7/HEADING DATE 2* (*HD2*); (**D**) *PRR3*; (**E**) *GIGANTEA* (*GI*); (**F**) *PHYTOCHROME A* (*PHYA*); (**G**) *EARLY FLOWERING 3a* (*ELF3a*)/*HD17*; (**H**) *ELF3b*. Red, high expression; blue, low expression.

**Figure 4 plants-12-00166-f004:**
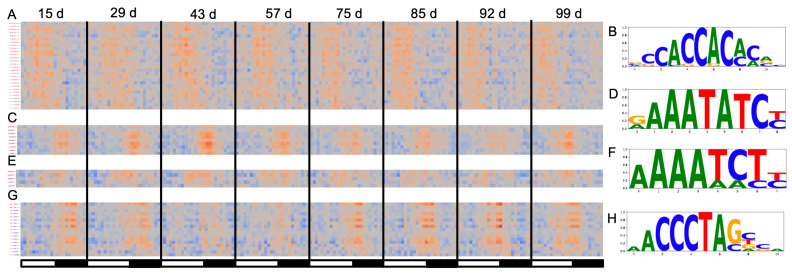
Element analysis across the developmental time course. (**A**) The Morning Element (ME; CCACAC) displayed morning-specific overrepresentation; (**B**) the logo plot of the significant ME 3–8 bp kmers found in (**A**); (**C**) the Evening Element (EE; AAATATCT) displayed evening/dusk-specific overrepresentation; (**D**) the logo plot of the significant ME 3–8 bp kmers found in (**C**); (**E**) the CCA1 Binding Site (CBS; AAAATCT) displayed evening/dusk-specific overrepresentation; (**F**) the logo plot of the significant ME 3–8 mers found in (**E**); (**G**) the Telobox (TBX; AAACCCT) displayed midnight-specific overrepresentation; (**H**) the logo plot of the significant ME 3–8 bp kmers found in (**G**). The white and black bars represent relative day and night, respectively.

**Figure 5 plants-12-00166-f005:**
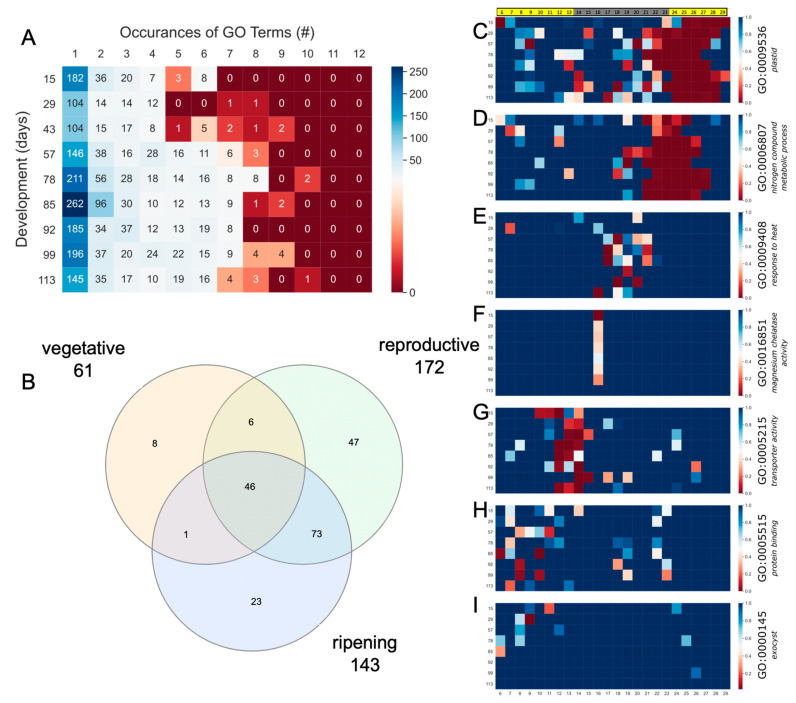
Gene ontology (GO) terms across the developmental time course. (**A**) Summary of the number of significant (*p* < 0.05) across the developmental time course and the number of times they are significant across each time course; (**B**) overlap of the GO terms across the vegetative, reproductive and ripening phases reveals 46 terms that are shared; significance plotted by TOD across the developmental time course for: (**C**) plastid (GO:0009532); (**D**) nitrogen compound metabolic process (GO:0006807); (**E**) response to heat (GO:0009408); (**F**) magnesium chelatase activity (GO:0016851); (**G**) transporter activity (GO:0005215); (**H**) protein binding (GO:0005515); (**I**) exocyst (GO:0000145). Green, more significant; red, less significant/non-significant.

**Table 1 plants-12-00166-t001:** Summary of the cycling genes across the rice developmental time courses.

	Leaf 15 d	Leaf 29 d	Leaf 43 d	Leaf 57 d	Leaf 78 d	Leaf 85 d	Leaf 92 d	Leaf 99 d	Leaf 113 d	Root 15 d	Root 43 d
total	27,648	27,648	27,648	27,648	27,648	27,648	27,648	27,648	27,648	27,648	27,648
Not expressed (#)	4735	4893	4713	4945	5180	5213	4552	5676	4701	1637	2265
Not Expressed (%)	17	18	17	18	19	19	16	21	17	6	8
Not rhythmic (#)	7298	9687	9618	5794	5473	2587	4862	4322	5354	23,652	24,167
Not rhythmic (%)	26	35	35	21	20	9	18	16	19	86	87
Rhythmic genes (#)	15,615	13,068	13,317	16,909	16,995	19,848	18,234	17,650	17,593	2359	1216
Rhythmic genes (%)	68	57	58	74	76	88	79	80	77	9	5

## Data Availability

All data are available in the supplement or upon request.

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
