# Peer review of "Time of Day Analysis over a Field Grown Developmental Time Course in Rice"

_plants, 2022, doi:10.3390/plants12010166_

Round 1
Reviewer 1 Report
Plants integrate time of day (TOD) information over an entire season to ensure optimal growth, flowering time, and grain ripening. However, most TOD expression studies have focused on a limited number of combinations of daylength and temperature under laboratory conditions. In this study, they re-analyze an Oryza sativa (rice) expression study that followed TOD expression in the field over an entire growing season and get some significant results. These results will provide a resource for researchers to study how TOD expression changes under natural conditions over a developmental time course, which will be important for researcher to engineer more resilient and prolific crops.
In general, the paper was well-written and data are well-present. Since all the data are from website, thus, there were no RT-PCR to confirm these results, which is the only weak point for this paper.
Author Response
Plants integrate time of day (TOD) information over an entire season to ensure optimal growth, flowering time, and grain ripening. However, most TOD expression studies have focused on a limited number of combinations of daylength and temperature under laboratory conditions. In this study, they re-analyze an Oryza sativa (rice) expression study that followed TOD expression in the field over an entire growing season and get some significant results. These results will provide a resource for researchers to study how TOD expression changes under natural conditions over a developmental time course, which will be important for researcher to engineer more resilient and prolific crops.
In general, the paper was well-written and data are well-present. Since all the data are from website, thus, there were no RT-PCR to confirm these results, which is the only weak point for this paper.
While specific expression patterns have not been confirmed by RT-PCR, extensive comparisons with other array data as well as RNA-seq have confirmed broad patterns of expression.
Reviewer 2 Report
Plants integrate time-of-day (TOD) information over seasons to optimize growth, flowering time, and grain filling. A genome-wide gene expression generated by the interaction between the circadian clock and environmental signals is considered to essential role for this process. However, transcriptome analyses focusing on TOD had been performed under laboratory conditions. In this manuscript, the author re-analyzed previously published transcriptome data in rice grown in the field, from vegetative to ripening growth phases. The DIURNAL pipeline that was established by the research group including that author, enabled to capture entire rice diel rhythmic pattern of the transcriptome, generating TOD expression. Comparison among the TOD expression in different growth phases clarified growth phase-dependent TOD expression. Although most core clock genes expression were consistent TOD expression over the season, two ELF3 genes were changed likely due to temperature and light cues. I totally agree with the author’s claim that the manuscript will provide a resource for researchers to study how TOD expression changed under natural conditions over whole rice life.
The text is easy to follow, but some figures were difficult to see because letters in figures were too small. I think this is from the figure preparation according to the MPDI format, but strongly recommend to modify the figures to see well.
Monocot PRR7/3 genes are annotated as PRR73 or PRR37, because it was difficult to distinguish PRR7 and PRR3 by sequence similarity (e.g., Murakami et al., Biosci. Biotechnol Biochem, 2007). Arabidopsis PRR3 is mutated at CCT motif that binds to DNA in vivo (Gendron et al., PNAS 2012, Nakamichi et al., PNAS 2012). I recommend to use proper names of PRR73 and PRR37 in rice (Figure 3, and text). Similarly, Sorghum ma1 was known to PRR37 (Murphy et al., PNAS 2011). Rice Hd2 was OsPRR37 (Koo et al., Mol Plant 2013).
Line 193. “timing of CAB expression (toc1)” should be “TIMING OF CAB EXPRESSION 1 (TOC1)”.
Author Response
Plants integrate time-of-day (TOD) information over seasons to optimize growth, flowering time, and grain filling. A genome-wide gene expression generated by the interaction between the circadian clock and environmental signals is considered to essential role for this process. However, transcriptome analyses focusing on TOD had been performed under laboratory conditions. In this manuscript, the author re-analyzed previously published transcriptome data in rice grown in the field, from vegetative to ripening growth phases. The DIURNAL pipeline that was established by the research group including that author, enabled to capture entire rice diel rhythmic pattern of the transcriptome, generating TOD expression. Comparison among the TOD expression in different growth phases clarified growth phase-dependent TOD expression. Although most core clock genes expression were consistent TOD expression over the season, two ELF3 genes were changed likely due to temperature and light cues. I totally agree with the author’s claim that the manuscript will provide a resource for researchers to study how TOD expression changed under natural conditions over whole rice life.
The text is easy to follow, but some figures were difficult to see because letters in figures were too small. I think this is from the figure preparation according to the MPDI format, but strongly recommend to modify the figures to see well.
The figures 1, 2, and 5 were updated to make sure all letters were readable.
Monocot PRR7/3 genes are annotated as PRR73 or PRR37, because it was difficult to distinguish PRR7 and PRR3 by sequence similarity (e.g., Murakami et al., Biosci. Biotechnol Biochem, 2007). Arabidopsis PRR3 is mutated at CCT motif that binds to DNA in vivo (Gendron et al., PNAS 2012, Nakamichi et al., PNAS 2012). I recommend to use proper names of PRR73 and PRR37 in rice (Figure 3, and text). Similarly, Sorghum ma1 was known to PRR37 (Murphy et al., PNAS 2011). Rice Hd2 was OsPRR37 (Koo et al., Mol Plant 2013).
Now that there are more complete (contiguous) genomes representing monocots and sister lineages of the angiosperms, it is now possible to show that PRR3 and PRR7 have distinct evolutionary trajectories (Michael Plant Phys 2022). This is important to point out since it is PRR3 in Sorghum that impacts flowering (ma1), while Hd2 is PRR7 in rice; this is not a statement about their specific function though, it is just based on evolutionary trajectory (syntenic orthology). The use of these more specific gene names is important because it is becoming clear that PRR3 is a specific target in other crops such as soybean.
Line 193. “timing of CAB expression (toc1)” should be “TIMING OF CAB EXPRESSION 1 (TOC1)”.
Corrected to add the 1, but this is the mutant so it is left lowercase.